# Analysis of Vulnerability on Weighted Power Networks under Line Breakdowns

**DOI:** 10.3390/e24101449

**Published:** 2022-10-11

**Authors:** Lixin Yang, Ziyu Gu, Yuanchen Dang, Peiyan He

**Affiliations:** School of Mathematics & Data Science, Shaanxi University of Science and Technology, Xi’an 710021, China

**Keywords:** complex network, cascading failure, vulnerability, redistribution rule

## Abstract

Vulnerability is a major concern for power networks. Malicious attacks have the potential to trigger cascading failures and large blackouts. The robustness of power networks against line failure has been of interest in the past several years. However, this scenario cannot cover weighted situations in the real world. This paper investigates the vulnerability of weighted power networks. Firstly, we propose a more practical capacity model to investigate the cascading failure of weighted power networks under different attack strategies. Results show that the smaller threshold of the capacity parameter can enhance the vulnerability of weighted power networks. Furthermore, a weighted electrical cyber-physical interdependent network is developed to study the vulnerability and failure dynamics of the entire power network. We perform simulations in the IEEE 118 Bus case to evaluate the vulnerability under various coupling schemes and different attack strategies. Simulation results show that heavier loads increase the likelihood of blackouts and that different coupling strategies play a crucial role in the cascading failure performance.

## 1. Introduction

The electricity supply network is an essential part of the infrastructure of our modern society. A power network is a complex and highly interconnected network, consisting of generators, consumers, substations and transmission lines [1,2]. Furthermore, a power network must maintain its power distribution function even when a few elements are changed. Therefore, the robust operation of physical distribution across a power network is fundament for our daily life. In recent years, due to the environment and the economy, there exists a shift from using central power sources to using small power sources [3,4,5,6,7]. More and more renewable energies are becoming part of the power network in modern society. This leads to many challenges concerning a power system’s ability to maintain its function under various conditions. 

On the other hand, it has been noticed that transmission lines with large capacities can deliver heavier flows. Each line, however, has a finite capacity. In order for a line to function properly, its flow cannot be more than the capacity at all times. Therefore, when the power network is damaged by various attacks, for instance, the removal of some transmission lines, it may cause huge disturbances and cause a possible blackout [8,9,10,11,12]. Cascading failures in power networks are the main cause of large-scale blackouts, which cause substantial costs. For a power network, the dynamic cascading failure process can be regarded as a sequence of tripping events leading eventually to a power outage spreading over a very large area. Over the past few years, many researchers have investigated load models and analyzed cascading failures by the Kuramoto-like model [13,14,15,16]. Furthermore, most of this cascading failure model is related to the analysis of steady states because cascading failures often exhibit dynamical transients. In addition to this, in many real-world networks, there is a weight related to lines and nodes; a network is considered weighted when the larger the weight on a line or node, the harder this line or node is to fail. Therefore, it is natural that the study of cascading failures in weighted power systems has been a very important topic in power electrical engineering. Wang et al. [17] constructed a cascading failure model with edge overloading for a weighted network and proposed a local weighted flow redistribution rule. Then, in [18], Wu et al. studied cascading failures in the BA network by the load on each node. Recently, S. Muldoon [19] et al. generated weighted small-world networks. In order to capture the relationship between the weighted feature and the cascading failures, this paper intends to put forward a nonlinear weighted flow redistribution rule and construct a cascading failure model.

In earlier work, researchers studied the cascading failures of the single-layer complex network. In [20], the authors proposed three resilience reinforcement strategies based on the nodal capacity redundancy at different structure scales. The performance of the reinforcement strategy had a close correlation with the nodal capacity redundancy. Many approaches analyze the vulnerability of power systems by removing components from the system and evaluate the resultant impact under different removing strategies [21]. Some studies use the Monte Carlo method to simulate all possible contingencies so that the most likely failed components can be identified [22].

However, in many practical situations, interdependent networks are ubiquitous in modern society. With the development of information technologies and power networks, the traditional power network gradually evolved to become cyber-physical interdependent networks [23,24,25,26,27]. More complex coupling formats may increase the vulnerability of the power network. Therefore, it is vital to investigate the cascading failures of interdependent networks. Buldyrev et al. [28] proposed a framework to analyze the robustness of an interdependent network. Later, Dong et al. [29] studied partially interdependent networks. In the previous model, they neglected some concrete engineering characteristics [30,31,32]. It is more feasible to analyze the structural vulnerability of interdependent power networks when combining the electrical engineering features with the complex network theory. Moreover, we generalize the betweenness concept to construct more practical cyber-physical interdependent networks. Moreover, we combine the proposed model with different coupling strategies to analyze the structural vulnerability under line breakdowns.

In this paper, we study the vulnerability in a weighted power network. In Section 2, we present a nonlinear model to analyze the cascading failure of a weighted power network, including a single-layer network and an interdependent network. Following that, Section 3 investigates the influence of coupling schemes and the mean degree of vulnerability of an interdependent network. Furthermore, the method is illustrated by examples of several power networks. Finally, conclusions and future work are provided in Section 4.

## 2. Cascading Failure Model for a Weighted Power Network

In this section, we focus on some existing networks to analyze the cascading failure process. The studied network ensembles are the single and interconnected weighted networks. 

### 2.1. Single Weighted Power Network

In general, a power network is summarized as a directed and weighted network. Moreover, there is a weight associated with each link or node, the larger the weight on the link or node, the harder this line or node is to fail. The electrical distance between the generator and consumer is defined as the equivalent impedance, which considers the impedance of the transmission lines between them. Additionally, betweenness centrality is considered as the most representative of the topological properties. However, the pure topological concepts disregard the concrete engineering characteristics of a power network. Therefore, we refine the betweenness as the extended betweenness to construct the power network in this subsection. Based on the above-mentioned betweenness, a new nonlinear model is proposed for studying the vulnerability in the weighted power network with different topologies.

According to the electrical circuit theory, the contribution of the transmission line to the power transmission can be assessed using the power transfer distribution factors (PTDF). PTDF can be denoted by an $N_{L}$$\times N_{B}$ matrix F, where D,G,L denote the set of consumer nodes, generator nodes and transmission lines; $N_{L}=\dim(L),N_{B}=\dim(G),$ respectively. The element $f_{ej}$ of the matrix express the change of power on each line e for a unit change of power injected at node i and delivered at the reference node. $f_{e}{}^{gd}$ represents the change of power on line e that is supplied at generator g and demanded at consumer d, and can be computed as follows:(1)$$f_{e}{}^{gd}=f_{eg}-f_{ed},e\in L$$

On the other hand, the power transmission line capacity is given by:(2)$$C_{g}^{d}=\underset{e\in L}{\min}({P_{e}{}^{\max}/\left|f_{e}\right|}^{gd})$$
where $P_{e}{}^{\max}$ is the transmission limit of line e. 

Based on the above specific characteristics of a power network, the extended betweenness of a node can be refined as:(3)$$T(w)=\frac{1}{2}\sum_{g\in G}\sum_{d\in D}C_{g}^{d}\sum_{e\in L^{w}}\left|f_{e}{}^{gd}\right|,w\neq g\neq d\in B$$
where $\sum_{e\in L^{w}}\left|f_{e}{}^{gd}\right|$ represents the sum of the PTDFs of all lines connecting a node w when a unit of power supplied at node w and demanded at node d. 

It is obvious that the extended betweenness might be close to reality in a power network. Hence, the weight of the nodes in this network depends on their extended betweenness. 

First, we assume the node load *j* for weighted network, and the power distribution model is given by:(4)$$L_{j}=(1+\gamma )\left(s_{i}\sum_{j\in \Gamma_{i}}s_{j}\right)$$
where γ>0 represents the load parameter, $s_{i}$ denotes the weight of node i and $\Gamma_{j}$ is the collection of neighboring nodes. Moreover, each node j has a threshold value, which is defined as:(5)$$C_{j}=L_{j}+\alpha L_{j}{}^{\beta }$$
where $\alpha \in \left[0,1\right]$,β
≥0 denote capacity parameters. 

If a node fails, the neighboring nodes will receive the loads in proportion to its remaining capacity $\pi_{j}=\frac{C_{j}-L_{j}}{\sum_{j\in \Gamma_{i}}\left(C_{j}-L_{j}\right)}$, and the received additional load for node *j* from node *i* can be described as:(6)$$\Delta L_{ij}=L_{i}\times \pi_{j}==(1+\gamma )(s_{i}\sum_{j\in \Gamma_{i}}s_{j})\times \frac{{\left(s_{j}\sum_{n\in \Gamma_{j}}s_{n}\right)}^{\beta }}{\sum_{j\in \Gamma_{i}}{\left(s_{j}\sum_{n\in \Gamma_{j}}s_{n}\right)}^{\beta }}$$
when $L_{j}+\Delta L_{ij}>C_{j}$, the neighboring node j will also fail, and its load will be further distributed to its neighboring nodes, which may cause the failure of neighboring nodes and form a cascading failure. Only if $L_{j}+\Delta L_{ij}\leq C_{j}$, does node j not fail, and thus the cascade failure stop. 

Based on the above analysis, it is obvious that the capacity parameters play a central role in the cascading failure of a power network. Thus, we define the threshold value of capacity parameter as $\alpha^{*}$, which is the minimum value required to avoid a cascading failure. That is, all node flows must be less than their threshold for the cascading failure to stop. In order to avoid a cascading failure, the inequality $L_{j}+\Delta L_{ij}\leq C_{j}$ must be satisfied.

In what follows, we substitute the load and capacity formula into the inequality:(7)$$L_{j}+L_{i}\times \frac{\alpha L_{j}{}^{\beta }}{\sum_{j\in \Gamma_{i}}\alpha L_{j}{}^{\beta }}\leq L_{j}+\alpha L_{j}{}^{\beta }$$

One can thus obtain:(8)$$\frac{(1+\gamma )(s_{i}\sum_{j\in \Gamma_{i}}s_{j}){\left(s_{j}\sum_{n\in \Gamma_{j}}s_{n}\right)}^{\beta }}{\sum_{j\in \Gamma_{i}}{\left(s_{j}\sum_{n\in \Gamma_{j}}s_{n}\right)}^{\beta }}\leq \alpha {\left(1+\gamma \right)}^{\beta }{\left(s_{j}\sum_{n\in \Gamma_{j}}s_{n}\right)}^{\beta }$$

We can further simplify and obtain:(9)$$\frac{(s_{i}\sum_{j\in \Gamma_{i}}s_{j}){\left(s_{j}\sum_{n\in \Gamma_{j}}s_{n}\right)}^{\beta -1}}{\sum_{j\in \Gamma_{i}}{\left(s_{j}\sum_{n\in \Gamma_{j}}s_{n}\right)}^{\beta }}\leq \alpha {\left(1+\gamma \right)}^{\beta -1}{\left(s_{j}\sum_{n\in \Gamma_{j}}s_{n}\right)}^{\beta -1}$$

To explore the relationship between the capacity parameters, we define the conditional probability that the neighboring node of the node with power $s^{\prime }$ is $P(\left.s^{\prime }\right|s_{i})$.

According to Bayesian formula, one can get:(10)$$\sum_{j\in \Gamma_{i}}s_{j}=\sum_{{}^{s^{\prime }=s_{\min}}}^{s_{\max}}s_{i}P(\left.s^{\prime }\right|s_{i})s^{\prime }$$

Additionally, the conditional probability satisfies the following equation:(11)$$P(\left.s^{\prime }\right|s_{i})=s^{\prime }P(s^{\prime })/\langle s\rangle$$

Substituting Equations (10) and (11) into (9), we can see that:(12)$$\frac{s_{i}s_{j}{}^{2\beta -2}\langle s\rangle }{\langle s^{2\beta +1}\rangle }\leq \alpha {\left(1+\gamma \right)}^{\beta -1}\times \frac{s_{j}{}^{2\beta -2}{\langle s^{2}\rangle }^{\beta -1}}{{\langle s\rangle }^{\beta -1}}$$

Furthermore, Equation (12) can be simplified as:(13)$$\alpha \geq \frac{s_{i}{\langle s\rangle }^{\beta }}{{\left(1+\gamma \right)}^{\beta -1}{\langle s^{2}\rangle }^{\beta -1}\langle s^{2\beta +1}\rangle }$$

Then, the threshold $\alpha^{*}$ of the capacity parameter α is given by:(14)$$\alpha^{*}=\left\{\begin{array}{l} \frac{s_{\max}\langle s\rangle }{\langle s^{3}\rangle },\beta =1\\ \frac{s_{\max}\langle s\rangle \langle s^{2}\rangle }{{\left(1+\gamma \right)}^{\beta -1}\langle s^{\beta +1}\rangle \langle s^{\beta +2}\rangle },\beta \neq 1 \end{array}\right.$$

In order to further analyze the effect of the capacity parameter. We let:(15)$$Z=\frac{s_{\max}\langle s\rangle \langle s^{2}\rangle {\left(1+\gamma \right)}^{1-\beta }}{\langle s^{\beta +1}\rangle \langle s^{\beta +2}\rangle }$$

We take the derivative of function (20) with respect to parameter β: (16)$$Z^{\prime }(\beta )=\frac{-N^{4}s_{\max}\langle s^{2}\rangle \langle s\rangle {\left(1+\gamma \right)}^{1-\beta }\ln(1+\gamma )}{{\left(\sum_{i=1}^{N}s_{i}{}^{2\beta +3}\right)}^{2}}-\frac{N^{2}(2\beta +3)\sum_{i=1}^{N}s_{i}{}^{2\beta +2}s_{\max}\langle s^{2}\rangle \langle s\rangle {\left(1+\gamma \right)}^{1-\beta }}{{\left(\sum_{i=1}^{N}s_{i}{}^{2\beta +3}\right)}^{2}}$$
because of β≥0, the inequality $Z^{\prime }(\beta )<0$ always holds, and Z(β) is a decreasing function.

In addition, we take the derivative of function (16) with respect to parameter γ, one can get:(17)$$Z^{\prime }(\gamma )=\frac{s_{\max}\langle s\rangle \langle s^{2}\rangle }{\langle s^{\beta +2}\rangle \langle s^{\beta +1}\rangle }\times (1-\beta ){\left(1+\gamma \right)}^{-\beta }$$

From Equation (17), one can find that when β>1, then function $Z^{\prime }(\gamma )<0$. Hence, function Z(γ) is a monotone minus function. Interestingly, it is noted that when parameter β>1, the higher load γ and the smaller $\alpha^{*}$; however, when 0<β<1, the smaller load γ, the smaller $\alpha^{*}$. That is, a smaller load parameter γ and a larger capacity parameter *β* enhances the robustness of the network substantially. The following numerical simulations show the validity of theoretical analysis.

Figure 1 illustrates parameters β,γ as a function of the threshold parameter α, and it is found that the lower the value of parameter γ, the stronger the robustness of the weighted power network with parameter β>1. Nevertheless, it is observed that parameter $\alpha^{*}$ increases with the parameter *γ* when 0<β<1. Thus, the simulation results are consistent with the theoretical results.

To further understand the different cascade responses in more detail, the relative size of the maximum connected subgraph was used to quantify the robustness of the weighted power network.
(18)$$G=N^{\prime }/N$$
where $N^{\prime }$ and N are the number of nodes of the maximum connected sub-network before and after a cascading failure, respectively.

Take the IEEE118 system as an example [33], we investigated the cascading failure process and compared two different attack patterns for a given power network. Before we started the simulations, we supposed that the network was in its stable state. The topology structure of the IEEE system is illustrated in Figure 2.

We assumed that interdependent relationship and redistribution rules will cause the cascading failure at the same time. The attack on the power network is represented by a random removal of nodes. To explore the effect of a small initial attack on our cascading failure, we calculated S, which represents the number of disabled nodes in the network after the cascading process based on the interdependent relationship and redistribution rules are both over.

Firstly, we initiated the random attack process by removing 10 nodes, we investigated the influence of attack strategies on the cascading failure process. The betweenness value of components was ranked. Vulnerability of power network was analyzed by progressively removing the nodes. More specifically, intentional attack and random attack strategies were considered. In the following simulation, the values of parameter were selected as α=0.5, β=1, γ=0.3.

From Figure 3, one can observe that the power network had stronger robustness under an intentional attack at the initial time. In contrast, the robustness of the power network was stronger under a random attack than an intentional attack with the evolution of time. For the sake of completeness, we also showed the evolution of the capacity parameter α with different attacks. Thus, we fixed the parameters β=1, γ=0.3, then adjusted the capacity parameter α to observe the performance of the power network. We simulated cascading failures in the weighted IEEE 118 system using the distribution rule proposed above.

From Figure 4, as expected, a larger capacity parameter resulted in fewer node failures, because it makes the overload condition more difficult to be satisfied. 

### 2.2. Multilayer Interdependent Network Model

As is known, interdependent networks are ubiquitous in modern society. With the development of information technologies, the traditional power system has evolved into an electrical cyber-physical power network. In addition, failures in one network may spread to its coupled networks. This section is devoted to the investigation of cascading failures for this kind of interdependent network. Furthermore, interdependence between a power system and its cyber network can enhance the vulnerability of the whole system, so it is vital to analyze the cascading failure of an interdependent network. 

According to the above-mentioned topological features, the extended betweenness was applied to IEEE118 system. Then, we constructed an electrical cyber network based on the IEEE118 system. Therefore, the power network and its cyber network can be described as: (19)$$G=\{V,E,M,W\},\;\left\{\begin{array}{l} V=\{V_{P},V_{C}\}\\ E=\{E_{P},E_{C}\}\\ M=\{M_{P},M_{C},M_{PC}\}\\ W=\{W_{P},W_{C}\} \end{array}\right.$$
where V,E are the set of nodes and lines, respectively. Additionally, $M_{P},M_{C}$ represent the adjacency matrix of the power network and its cyber network. $M_{PC}$ is the interlayer coupling matrix between two layers. $W_{C}$ and $W_{P}$ are the arrangement of weight. In addition, P,C denote the power network and cyber network, respectively. 

Specifically, the studied interdependent network can be considered as a partial one-to-one network as depicted in Figure 5, where the lower layer is the power network, the upper layer is its cyber network. Cyber network is in charge of controlling the power network while the power network provides electricity to its cyber network. In addition, the number of nodes in the cyber network is larger than that in the power network because of the existence of autonomous nodes. Hence, it is important to analyze the influence of coupling strategies on the vulnerability of an electrical cyber-physical interdependent network. In what follows, we analyze the performance of interdependent network with two different coupling schemes.

## 3. Vulnerability Analysis for a Power Network

### 3.1. The Influence of Coupling Strategy on Vulnerability

This subsection is devoted to the investigation of vulnerability for the electrical cyber-physical network. Next, to evaluate the vulnerability of a power network under different attacks, we define the normalized avalanche size as:(20)$$S=\frac{{N_{C}}^{\prime }+{N_{P}}^{\prime }}{N_{C}+N_{P}}$$
where $N_{C},N_{P}$ denote the number of survival nodes in the cyber network and the power network before an attack. ${N_{C}}^{\prime }$ and ${N_{P}}^{\prime }$ are the number of survival nodes in the cyber network and the power network after an attack. In fact, the degree is equal to the number of connectivity links a node is connected to and plays a crucial role in ensuring connectivity of the networks. We investigate the node failures as a function of the number of attacks. It is well known that the performance of an interdependent network can be influenced by the coupling patterns and attack strategies. The following coupling schemes are considered: (1) betweenness-extended betweenness (BT), and (2) degree-betweenness (DT).

In next step, a scale-free network with a mean degree $\langle K\rangle =3$ is constructed based on IEEE118 systems, and construct an electrical cyber-physical network via BT and DT coupling schemes. If we neglect autonomous nodes in the cyber network, then each node in a layer mutually depends on only one node in another layer with one-to-one matching. Furthermore, the performance of the interdependent electrical cyber-physical network is shown according to different attacks. Here, we focus on intentional and random attacks.

One can find that inter-coupling strategies can influence the network’s performance from Figure 6. There are negligible differences for two coupling strategies under random attacks. Nevertheless, the electrical cyber-physical interdependent network undergoes second-order transition under intentional attacks. At the initial time, the BT-coupling strategy has the stronger robustness than the relationship based on DT-coupling under the same conditions.

### 3.2. The Influence of Mean Degree on Vulnerability

On the other hand, if the topology of the weighted power network is selected as the IEEE 118 system; therefore, the topology structure of the power network is fixed. Thus, the topology structure of its cyber network plays a major role in the performance of the interdependent network. Especially, the mean degree of a node plays a crucial role in the cascading failure process. In this subsection, we investigate the role of the mean degree on vulnerability of an electrical cyber-physical interdependent network. Here, the same homogeneous coupling and distribution of generators and consumers was adopted, as in Figure 7.

Figure 7a depicts the cascading failure process of the cyber network with different mean degrees under random attacks. Figure 7b presents the same case under intentional attacks. In Figure 7b, we can see similar results to Figure 7a, which shows S as a function of the attack strategy. Moreover, Figure 7 reveals that the robustness of the interdependent network is improved with an increase in the average degree of the cyber network. However, at the same time, after some nodes fail, the probability of the cyber sub-network enhances. This leads to the failure of the nodes of the power network, and the collapse of the interdependent network. It is shown that the network’s topology plays a significant role in determining the dynamics of cascading failures.

## 4. Conclusions

In summary, this paper addressed the vulnerability of weighted power networks. We present the extended betweenness to construct a more practical nonlinear model to analyze the cascading failure from edge overloading on weighted networks. Specially, the proposed model takes into consideration the contribution of the transmission line, moreover, the role of the capacity parameter on the cascading failure for a weighted power network. According to the presented weighted power network model, a weighted electrical cyber-physical interdependent network was developed. Furthermore, we analyzed the vulnerability and cascading failure dynamics on the entire interdependent network. Our results show that interdependent networks undergo second-order transition under intentional attacks and the robustness is improved with the increase of the average degree of the cyber network. These results indicate that the significant role of weights and the interdependent relationship in power networks for designing protective strategies against cascading failures.

Vulnerability control and optimization offer new avenues for controlling the dynamic behavior of real-world systems. Additionally, some artificial intelligence approaches can be used to obtain a general strategy for guiding the improvement of cascading failures for multiplex networks, which also deserves more attention in the future.

## Figures and Tables

**Figure 1 entropy-24-01449-f001:**
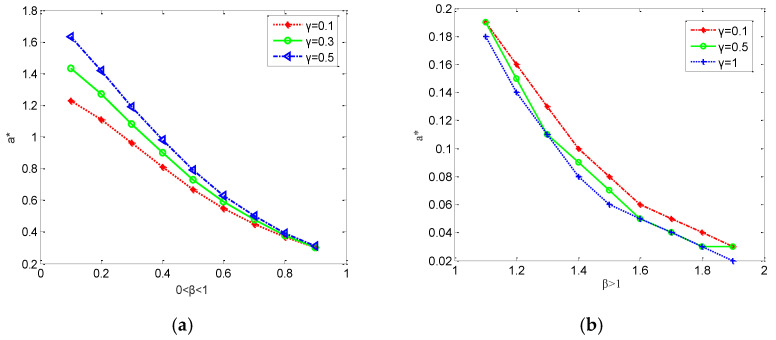
The relationship between threshold $\alpha^{*}$ and β,γ. (**a**) When parameter 0<β<1 (**b**) When parameter β>1.

**Figure 2 entropy-24-01449-f002:**
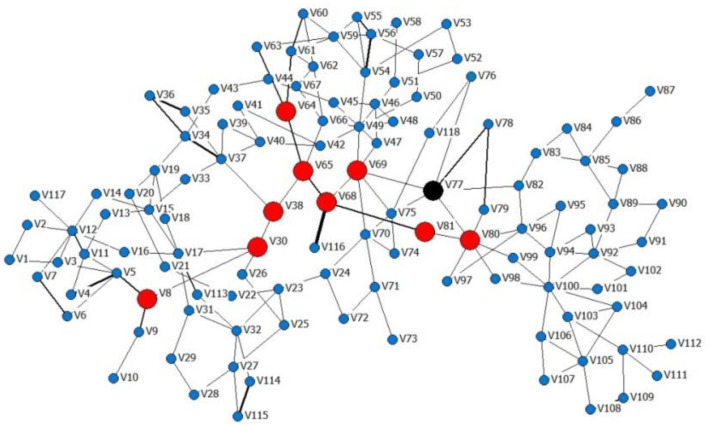
The topology of the weighted IEEE118 system.

**Figure 3 entropy-24-01449-f003:**
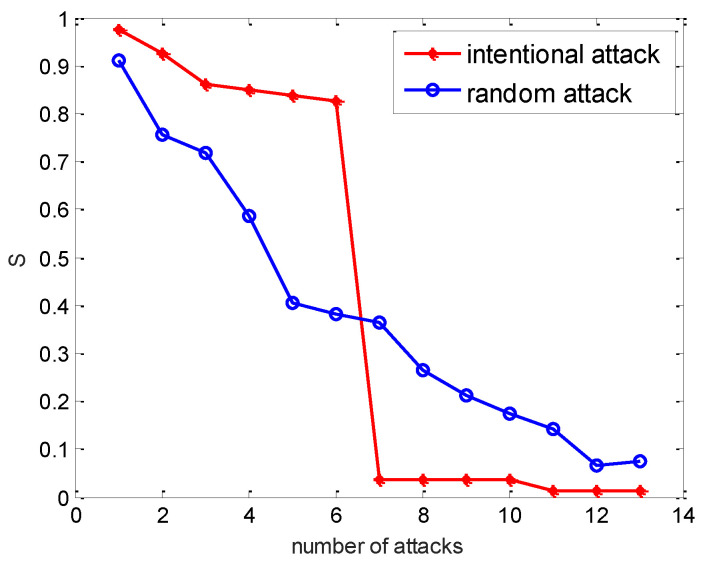
Relative size of the most connected sub-network of the IEEE118 systems under different attack strategies.

**Figure 4 entropy-24-01449-f004:**
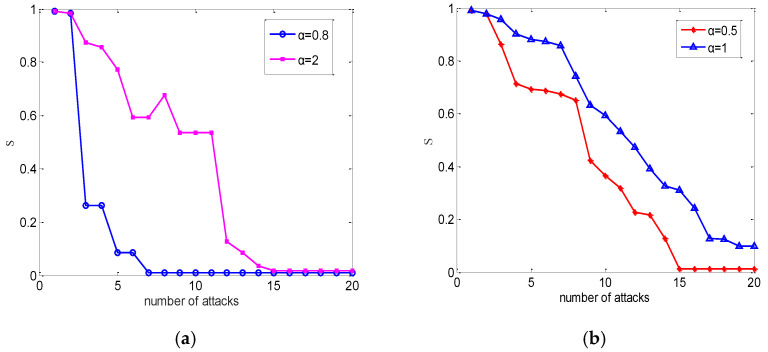
The case of a weighted power network under different attack strategies. We averaged over 50 realizations. (**a**) Random attack. (**b**) Intentional attack.

**Figure 5 entropy-24-01449-f005:**
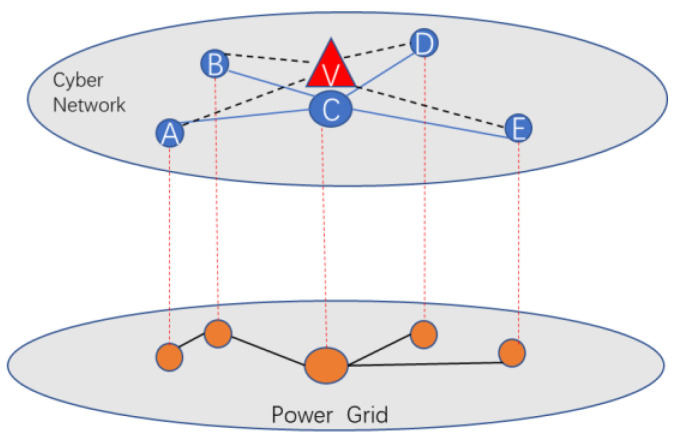
The topology of an electrical cyber-physical interdependent network.

**Figure 6 entropy-24-01449-f006:**
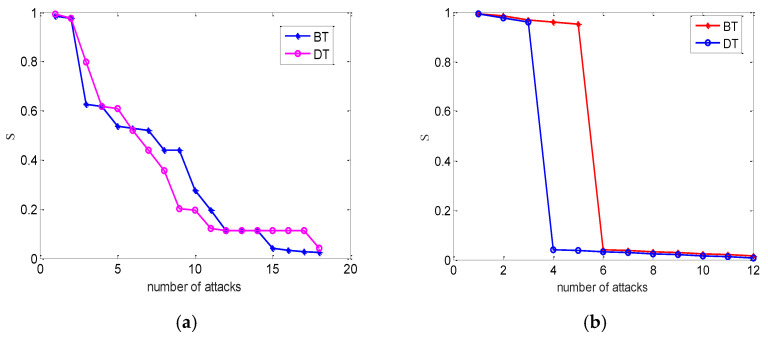
The case of the coupled network with partial one-to-one interdependent nodes of the power network and its cyber network with different coupling strategies. The numerical simulations with $N_{A}=118,N_{B}=136$ and mean degree $\langle k_{A}\rangle =\langle k_{B}\rangle$. (**a**) Random attack. (**b**) Intentional attack. We averaged over 30 realizations.

**Figure 7 entropy-24-01449-f007:**
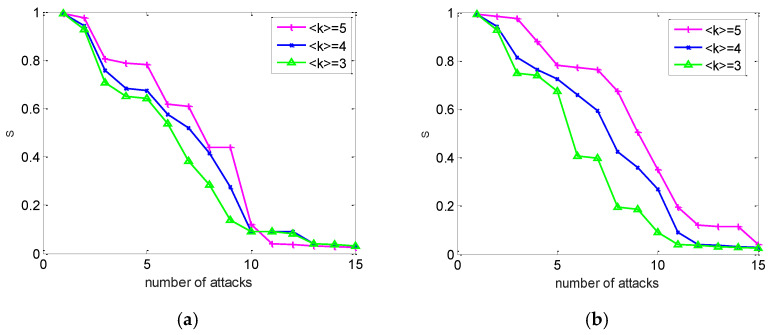
Influence of the average degree on the robustness of an interdependent network. (**a**) Random attack. (**b**) Intentional attack.

## Data Availability

The datasets generated and analyzed during the current study are available from the corresponding author on reasonable request.

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
