# Peer review of "Analysis of Vulnerability on Weighted Power Networks under Line Breakdowns"

_entropy, 2022, doi:10.3390/e24101449_

Round 1

Reviewer 1 Report

The analysis of vulnerability on weighted power network under line breakdowns is presented

in the research.

The abstract directly provides the information about the investigation of vulnerability of weighted power network along with the propose solution.

However the abstract should start from (i) the research problem, aims and objectives, then (ii) research methodology, (iii) findings and results and finally (iv) the conclusion along with the recommendations.

Five to sever keywords related to the research area should be used.

Introduction is written well, but can be improved further.

The organization of the article is presented at the end of the introduction section should include the section numbers.

The title of Section 2 should be revised. It seems that the analysis is performed first then a model is presented, however the related work is shown in this section.

It is suggested to present the model in the form of figure with explanation.

Please double check the headings of the manuscript and match them with the research topic.

Conclusion is written well, but can be further improved along with the future directions.

Author Response

Reviewer #1: The abstract directly provides the information about the investigation of vulnerability of weighted power network along with the propose solution. However the abstract should start from (i) the research problem, aims and objectives, then (ii) research methodology, (iii) findings and results and finally (iv) the conclusion along with the recommendations.

Answer: According to the reviewer’s comment, we have revised the abstract.

  1. Five to sever keywords related to the research area should be used.

Answer: According to the reviewer’s comment, we have revised the abstract.

  1. Introduction is written well, but can be improved further.

Answer: According to the reviewer’s comment, we have revised the introduction.

  1. The organization of the article is presented at the end of the introduction section should include the section numbers.

Answer: According to the reviewer’s comments, we have added the section numbers.

  1. The title of Section 2 should be revised. It seems that the analysis is performed first then a model is presented, however the related work is shown in this section.

Answer: Considering the reviewer’s suggestion, we have revised the title of Section 2.

  1. It is suggested to present the model in the form of figure with explanation.

Answer: It is really true as reviewer suggested that the figures can really help the reader. But we would like to stress that it is difficult to present the model in the form of figures. On the one hand, the model has many parameters. On the other hand, the form of model is complex.

  1. Please double check the headings of the manuscript and match them with the research topic.

Answer: Considering the reviewer’s suggestion, we have revised the headings of the manuscript.

  1. Conclusion is written well, but can be further improved along with the future directions.

Answer: Considering the reviewer’s suggestion, we have added the future directions in Conclusion Section.

Reviewer 2 Report

The authors of this article present analyzes related to the vulnerability of the weighted power network to failures. They propose their own approach to testing the vulnerability and resilience of the loaded network. The method used is to increase efficiency and effectiveness. The article emphasizes the need to perform such analyzes from the point of view of improving the reliability and security of the network operation. However, there are some considerations and questions that should be addressed and clarified:

• In my opinion, the literature review is too cursory on the topic presented. Authors should review existing articles in more detail.

• Did the authors perform such analyzes for larger networks (with a greater number of nodes), eg networks with several thousand nodes. What is the effectiveness of the proposed method for such networks? What is the computation time? Does the size (size) of the network affect these parameters?

• Schema and data of the IEEE 118 network - no reference to the literature is available (data source). In addition, there is no general description and characteristics of the test network.

• Authors should describe and explain their analysis and calculation assumptions in more detail. Certain mental shortcuts are used and the reader must make some guesswork. Eg page 6 and 7 - 10 nodes removed and impact on cascading failures assessed? Please write why these nodes were selected and what are the failures? Will selecting more nodes affect the effectiveness of the method? Each calculation and assumptions to them should be preceded by an appropriate detailed explanation.

• The numbering of the figures is incorrect - Fig 4 is first and then Fig 1 (not consecutive).

• In my opinion, the font of the text in the graphs / figures is too small (the text in the figures is hardly legible), e.g. Fig. 4, Fig. 1, Fig. 2, Fig. 3, Fig. 5, Fig. 6.

• The single letters "a" or the articles "the" appear at the end of some lines. They should appear at the beginning of the next line.

• The conclusions should describe in detail what new the presented article brings in comparison with other works of this type.

Author Response

Reviewer #2: The authors of this article present analyzes related to the vulnerability of the weighted power network to failures. They propose their own approach to testing the vulnerability and resilience of the loaded network. The method used is to increase efficiency and effectiveness. The article emphasizes the need to perform such analyzes from the point of view of improving the reliability and security of the network operation. However, there are some considerations and questions that should be addressed and clarified:

  1. In my opinion, the literature review is too cursory on the topic presented. Authors should review existing articles in more detail.

Answer: Considering the reviewer’s suggestion, the according contents have added in the revised paper.

  1. Did the authors perform such analyzes for larger networks (with a greater number of nodes), eg networks with several thousand nodes. What is the effectiveness of the proposed method for such networks? What is the computation time? Does the size (size) of the network affect these parameters?

Answer: In fact, we haven’t performed such analyzes for larger networks with several thousand nodes. From a theoretical point of view, the proposed model is effectiveness. However, the approaches of target attacks and topology structures of power network play crucial role on the process of cascading failure. Hence, the size of the network will affect the parameters.

  1. Schema and data of the IEEE 118 network - no reference to the literature is available (data source). In addition, there is no general description and characteristics of the test network.

Answer: Considering the reviewer’s suggestion, we have added the according reference.

The following reference gives the description of IEEE 118network in details.

[1] S. Peyghami, P. Davari, M. Fotuhi-Firuzabad, F. Blaabjerg, Standard test systems for modern power system analysis: An overview, in IEEE Industrial Electronics Magazine, 2019, 13:86-105.

  1. Authors should describe and explain their analysis and calculation assumptions in more detail. Certain mental shortcuts are used and the reader must make some guesswork. Eg page 6 and 7 - 10 nodes removed and impact on cascading failures assessed? Please write why these nodes were selected and what are the failures? Will selecting more nodes affect the effectiveness of the method? Each calculation and assumptions to them should be preceded by an appropriate detailed explanation.

Answer: We assume that interdependent relationship and redistribution rule will cause the cascading failure at the same time. The attack on the power network is represented by a random removal of nodes. To explore the effect of a small initial attack on our cascading failure, we calculate, which means the number of disabled nodes in network after the cascading process based on interdependent relationship and redistribution rule are both over.

  1. The numbering of the figures is incorrect - Fig 4 is first and then Fig 1 (not consecutive).

Answer: According to the reviewer’s comment, we have corrected the numbering of the figures.

  1. In my opinion, the font of the text in the graphs / figures is too small (the text in the figures is hardly legible), e.g. Fig. 4, Fig. 1, Fig. 2, Fig. 3, Fig. 5, Fig. 6.

Answer: According to the reviewer’s comment, we have revised the font of the text in the figures.

  1. The single letters "a" or the articles "the" appear at the end of some lines. They should appear at the beginning of the next line.

Answer: Following the suggestion of the reviewer, we have revised the according contents.

  1. The conclusions should describe in detail what new the presented article brings in comparison with other works of this type.

Answer: According to the reviewer’s comment, we have revised the conclusion and added the differences between our manuscript and other papers.

Reviewer 3 Report

This manuscript analyzed the dynamics resilience and vulnerability in the weighted power network under transmission line breakdowns. A nonlinear model was used for the cascading failure of weighted power network, including single-layer network and interdependent network. The influence of coupling schemes and mean degree on vulnerability of interdependent network was shown. Several examples were chosen to show the methods. I went through the method, and it is solid. I recommend it for publication, but it has some typos to be corrected. I just list two but there are more.

1.     The indices in equ. 1 don’t match the explanations above such as fgc vs fgd.

2.     Page 3, “cricuit”-> “circuit”.

Author Response

Reviewer #3:

This manuscript analyzed the dynamics resilience and vulnerability in the weighted power network under transmission line breakdowns. A nonlinear model was used for the cascading failure of weighted power network, including single-layer network and interdependent network. The influence of coupling schemes and mean degree on vulnerability of interdependent network was shown. Several examples were chosen to show the methods. I went through the method, and it is solid. I recommend it for publication, but it has some typos to be corrected. I just list two but there are more.

  1. The indices in equ. 1 don’t match the explanations above such as fgcvs fgd.
  2. Page 3, “cricuit”-> “circuit”.

Answer: Thanks the reviewer for pointing our mistake. We are sorry for our careless. 

Round 2

Reviewer 1 Report

Significant improvements have been observed in the revised manuscript, hence it is recommended for publication. 

Reviewer 2 Report

Thank you for your responses. Good luck